# Metabolic pathway alterations in microvascular endothelial cells in response to hypoxia

**Emily B. Cohen**[1], **Renee C. Geck**[1], **Alex Toker**[1,2]*

**1** Department of Pathology and Cancer Center, Beth Israel Deaconess Medical Center, Harvard Medical School, Boston, Massachusetts, United States of America, **2** Ludwig Center at Harvard, Harvard Medical School, Boston, Massachusetts, United States of America

* atoker@bidmc.harvard.edu

## Abstract

The vasculature within a tumor is highly disordered both structurally and functionally. Endothelial cells that comprise the vasculature are poorly connected causing vessel leakage and exposing the endothelium to a hypoxic microenvironment. Therefore, most anti-angiogenic therapies are generally inefficient and result in acquired resistance to increased hypoxia due to elimination of the vasculature. Recent studies have explored the efficacy of targeting metabolic pathways in tumor cells in combination with anti-angiogenic therapy. However, the metabolic alterations of endothelial cells in response to hypoxia have been relatively unexplored. Here, we measured polar metabolite levels in microvascular endothelial cells exposed to short- and long-term hypoxia with the goal of identifying metabolic vulnerabilities that can be targeted to normalize tumor vasculature and improve drug delivery. We found that many amino acid-related metabolites were altered by hypoxia exposure, especially within alanine-aspartate-glutamate, serine-threonine, and cysteine-methionine metabolism. Additionally, there were significant changes in *de novo* pyrimidine synthesis as well as glutathione and taurine metabolism. These results provide key insights into the metabolic alterations that occur in endothelial cells in response to hypoxia, which serve as a foundation for future studies to develop therapies that lead to vessel normalization and more efficient drug delivery.

## Introduction

The tumor vasculature is highly disordered with both dense regions of vessels and areas that lack vessels. This leads to inefficient blood flow and delivery of nutrients to a tumor, resulting in a microenvironment that is nutrient-poor and hypoxic [1–3]. The endothelial cells that comprise disordered vessels are not tightly connected, resulting in leaky vessels [4,5] and exposing endothelial cells to a hypoxic environment. This leakiness also facilitates tumor cell intravasation and metastasis and significantly impedes efficient drug delivery to tumors

**Data Availability Statement:** All relevant data are within the paper and its Supporting Information files.

**Funding:** E.B.C. was supported by a training grant from the National Heart Lung and Blood Institute (T32H007893; www.nhlbi.nih.gov). R.C.G. was supported by an individual National Research Service Award from the National Cancer Institute (F31CA213460; www.cancer.gov). The funders had no role in study design, data collection and analysis, decision to publish, or preparation of the manuscript. The specific roles of these authors are articulated in the 'author contributions' section.

**Competing interests:** A.T. is a paid consultant of and received support via salary from Oncologie, Inc. and Bertis, Inc. This does not alter our adherence to PLOS ONE policies on sharing data and materials.

[1,3,6–9]. Therefore, a number of therapeutic approaches in cancer treatment have focused on normalizing the disordered tumor vasculature.

Traditionally, the goal of anti-angiogenic therapies (AATs) was to eradicate the vasculature and deprive tumors of nutrients [10]. Many AATs are approved or in clinical trials for treatment of solid tumors as monotherapies or in combination with chemotherapy [9,11–15]. However, tumors adapt to the hypoxic microenvironment induced by AATs and develop resistance to therapy [16–22]. To overcome these limitations, an alternative approach to targeting the tumor vasculature has been explored, whereby pruning and normalization of the tumor vasculature promotes the highly ordered vasculature found within normal tissues. The resulting increased oxygenation and blood flow in turn allows efficient drug delivery [1,3,23]. Normalization of the tumor vasculature has been achieved in several clinical studies for various types of tumors [9,15,24,25]. However, studies have also shown that there is a narrow window in which to deliver normalization therapy before the vasculature is depleted, and hypoxia and drug resistance ensue [1,23,26].

Identification of targetable metabolic changes in tumor cells has also led to promising results in pre-clinical settings. Exploring reprogramming of metabolic enzymes and pathways in specific tumor types has identified unique metabolic requirements that can be targeted to restrict the enhanced proliferative and survival capacity of tumor cells whilst sparing normal cells [27–29]. Although many studies have focused on metabolic reprogramming of tumor cells, alterations of metabolic processes in the endothelial cells that comprise the tumor vasculature have not been explored in depth [30–33]. One key feature of the altered metabolism of endothelial cells is increased glycolytic flux and greater dependency on glycolysis [34,35]. For example, inhibition of the glycolytic activator, 6-phosphofructo-2-kinase/fructose-2,6-bisphosphatase 3 (PFKFB3), normalizes tumor vasculature by tightening endothelial cell junctions [34]. In addition to clinical trials for PFKFB3 inhibition, additional studies have explored inhibition of the fatty acid oxidation enzyme, carnitine palmitoyltransferase, and the glutamine and aspartate/asparagine metabolizing enzymes, glutaminase and asparagine synthetase, respectively [36].

Identifying novel metabolic alterations in endothelial cells in response to specific treatments or conditions such as hypoxia could provide information for the development of therapies to normalize the tumor vasculature and in turn, facilitate efficient delivery of tumor-targeting therapies [1,2]. These therapies could also be applied to correcting aberrant angiogenesis in diverse pathophysiologies beyond the tumor vasculature. Here, we explored the reprogramming of metabolic pathways in microvascular endothelial cells exposed to hypoxia using targeted metabolomics, in order to identify novel alterations that could be targeted to induce vessel normalization.

## Materials and methods

### Cell culture

HMEC-1 cells were a gift from the Tim Hla Laboratory (Boston Children's Hospital; May 2017) and were originally obtained from the American Type Culture Collection (ATCC; #CRL-3243), and MDA-MB-468 and 293T cells were obtained from the ATCC (#HTB-132; #CRL-3216) and authenticated using short tandem repeat profiling. None of the cell lines are listed in the ICLAC Database of Cross-contaminated or Misidentified Cell Lines. HMEC-1 cells were cultured in MCDB 131 medium (Wisent BioProducts) supplemented with 10 ng/mL epidermal growth factor (EGF) (R&D Systems), 1 μg/mL hydrocortisone (Sigma), 1 mM glutamine (Corning), and 10% heat-inactivated fetal bovine serum (HI-FBS) (Gibco) at 37°C in the presence of 5% carbon dioxide ($CO_2$). MDA-MB-468 and 293T cells were cultured in

DMEM (Wisent BioProducts) supplemented with 10% HI-FBS (Gibco) at 37˚C in the presence of 5% $CO_2$. Cells were passaged for no more than one month or eight passages. For hypoxia studies, hypoxia chamber (BioSpherix ProOx P110) was set to 1% oxygen and maintained at that level with gas mixture of 5% $CO_2$ and balance of nitrogen (LifeGas).

## Antibodies

Rabbit monoclonal HIF1α antibody (clone D1S7W; #36169; 1:1000; epitope: residues surrounding Leu478 of human HIF1α protein), rabbit monoclonal β-actin antibody (clone 13E5; #4970, 1:3000; epitope: residues near amino terminus of human β-actin protein), rabbit monoclonal xCT antibody (clone D2M7A; #12691, 1:1000; epitope: residues surrounding Ala224 of human xCT protein), and rabbit monoclonal vinculin antibody (clone E1E9V; #13901, 1:1000; epitope: amino terminus of human vinculin protein) were purchased from Cell Signaling Technology.

## Metabolites

N-acetyl-L-cysteine, L-aspartic acid, L-cysteine, dimethyl α-ketoglutarate (AKG), glutathione ethyl ester, L-homocystine, L-methionine, sodium α-ketobutyrate (AKB), and sodium pyruvate were purchased from Sigma. U-$^{13}$C$_3$$^{15}$N-L-cysteine was purchased from Cambridge Isotope Laboratories.

## Plasmids

pMXS-SLC1A3 retroviral expression vector was purchased from Addgene (#72873). pHAGE-xCT-HA-FLAG lentiviral expression vector was previously published [37]. pMSCV retroviral expression vector and pCL-Eco retroviral packaging vector were gifts from the DiMaio Laboratory (Yale University). All sequences were verified by DNA sequencing.

## Lentivirus transduction

Vesicular stomatitis virus (VSV)-G protein pseudotyped lentiviruses and retroviruses were prepared by using polyethylenimine (PEI; Sigma) to co-transfect 293T cells with a lentiviral or retroviral plasmid and pantropic VSVg (Addgene) and either psPAX2 (Addgene) (lentivirus) or pCL-Eco (retrovirus) packaging plasmids. After culture in HMEC-1 medium for 48 hrs at 37˚C, the viral supernatant was harvested, filtered through a 0.45 μm filter (Thermo Fisher Scientific), and used immediately. Cells stably expressing pHAGE-xCT-HA-FLAG were selected in medium containing 1 μg/mL puromycin dihydrochloride (Corning), and cells expressing pMXS-SLC1A3 were selected in 5 μg/mL blasticidin (Invivogen).

## Immunoblotting

Cells were washed with ice-cold PBS (Boston BioProducts) and lysed in radioimmuno-precipitation buffer (1% NP-40, 0.5% sodium deoxycholate, 150 mM Tris-hydrochloride (pH 7.5), 150 mM sodium chloride, 0.1% sodium dodecyl sulfate (SDS), protease inhibitor cocktail, 50 nM calyculin A, 20 mM sodium fluoride, 1 mM sodium pyrophosphate) for 15 min at 4˚C. Cell lysates were cleared by centrifugation at 14,000 rpm at 4˚C for 10 min, and protein concentration was measured by Bio-Rad DC protein assay. Lysates were resolved on polyacrylamide gels by SDS-polyacrylamide gel electrophoresis (SDS-PAGE) and transferred electrophoretically to a nitrocellulose membrane (Bio-Rad) at 100V for 100 min. xCT and vinculin membranes were blocked in tris-buffered saline with 1% tween (TBST) buffer (Boston BioProducts) containing 5% w/v nonfat dry milk (Andwin Scientific) for 1 hr and

then incubated with primary antibody diluted in TBST with 5% bovine serum albumin (BSA; Boston BioProducts) overnight at 4˚C. Membranes were washed three times with TBST and then incubated with horseradish peroxidase-conjugated secondary antibody (Millipore) for 1 hr at room temperature. Membranes were washed three times with TBST and developed using Clarity Western ECL Substrate (Bio-Rad) and Clarity Max Western ECL Substrate (Bio-Rad) and imaged on a Bio-Rad ChemiDoc imager.

For HIF1α and β-actin blots: immediately after cells were removed from incubator, media was aspirated and SDS sample buffer heated to 95˚C was added. Lysates were scraped from the plate, transferred to a microfuge tube, and pulse-sonicated. Lysates were resolved on a poly-acrylamide gel by SDS-PAGE and transferred electrophoretically to a nitrocellulose membrane (Bio-Rad). Membranes were blocked in TBS buffer (no tween; Boston BioProducts) containing 5% (w/v) nonfat dry milk for 1 hr and then incubated with primary antibody diluted in TBST with 5% BSA overnight at 4˚C. Membranes were washed three times with TBST and then incubated with IRDye® 680CW-conjugated secondary antibody (LI-COR). Membranes were washed three times with TBST and scanned using a LI-COR Odyssey CLx imager.

## Quantitative real-time polymerase chain reaction

Total RNA was isolated with the NucleoSpin RNA Plus kit (Macherey-Nagel) according to the manufacturer's protocol. Reverse transcription was performed using the Quantitect Reverse Transcription Kit (Qiagen). cDNA was detected using Power Up SYBR Master Mix (Applied Biosystems). Quantitative real-time PCR was performed using a CFX384 Touch Real-Time PCR Detection System (Bio-Rad). See S3 Table for list of primers. *ASS1* primers were designed using Primer3 [38]. All other primers without citations from previous studies were from the Roche Universal Probe Library. PCR was carried out in technical triplicate. Quantification of mRNA expression was calculated by the ΔCT method with *18S* ribosomal RNA serving as the reference gene.

## LC/MS-MS metabolomics profiling

Cells were seeded in technical duplicate at $5 \times 10^5$ cells per 10-cm plate for each condition. Designated plates cultured in 1% oxygen for 48 hrs or 3 weeks. For 3-week experiments, cells in hypoxia had intermittent exposure to normoxia to maintain the cell culture. For metabolite extraction, medium was aspirated and ice-cold 80% (v/v) methanol was added. Cells and the metabolite-containing supernatants were collected. Insoluble material was pelleted by centrifugation at 20,000 g for 5 min. The resulting supernatant was evaporated under nitrogen gas. Samples were resuspended using 20 μL HPLC-grade water for mass spectrometry. Five microliters from each sample were injected and analyzed using a 5500 QTRAP hybrid triple quadrupole mass spectrometer (AB/SCIEX) coupled to a Prominence UFLC HPLC system (Shimadzu) with HILIC chromatography (Waters Amide XBridge) via selected reaction monitoring (SRM) with polarity switching. A total of 276 fragments representing 265 endogenous water-soluble metabolites were targeted for steady-state analyses. Electrospray source voltage was +4950 V in positive ion mode and −4500 V in negative ion mode. The dwell time was 3 ms per SRM transition [39]. Peak areas from the total ion current for each metabolite were integrated using MultiQuant v2.1.1 software (AB/SCIEX). Metabolite total ion counts were the integrated total ion current from a single SRM transition and normalized by cellular protein content. Nonhierarchical clustering was performed by Metaboanalyst 4.0 using a Euclidean distance measure and Ward's clustering algorithm, and most significant metabolites determined by t-test / ANOVA [40].

## LC/MS-MS with $^{13}C_3^{15}N$-cysteine labeling

Cells were seeded in technical duplicate at $5x10^5$ cells per 10-cm plate for each condition. Designated plates were cultured in 1% oxygen for 48 hrs. Sixteen hours prior to end of the time period, medium was changed on all plates to either unlabeled or labeled media made from DMEM medium lacking glutamine, methionine, and cysteine (Wisent BioProducts) supplemented with glutamine, EGF, and hydrocortisone as above, along with 10% dialyzed FBS (Gibco) and 1 mM methionine. One hundred sixty-five micromolar cysteine was supplemented (Sigma) to negative controls for method validation; 165 µM U-$^{13}C_3^{15}N$-cysteine was supplemented for all experimental samples. Metabolite extraction was performed as described above. Five microliters from each sample (20 µL) were injected with similar methodology as above using a 6500 QTRAP (AB/SCIEX) and integrated using MultiQuant v3.0 software [41].

## Metabolic pathway analysis

Pathway Analysis was performed using Metaboanalyst 4.0 [40]. Parameters were defined as follows: Pathway Enrichment Analysis–Global Test, Pathway Topology Analysis–Relative-Betweenness Centrality, Pathway Library–KEGG *Homo sapiens* Oct2019.

## Sulforhodamine B cell proliferation assays

HMEC-1 cells or HMEC-1 cells overexpressing xCT or SLC1A3 were seeded into 24-well plates (Greiner Bio-One) at a density of $6.5 \times 10^3$ cells per well in 0.5 mL medium. The following day, medium was changed to 1 mL HMEC-1 growth medium with any indicated supplements. For assays with added aspartate, medium was changed to custom DMEM medium without glucose, glutamine, or pantothenate (US Biological) supplemented with 5.55 mM glucose (Gibco), 1 mM pyruvate (Sigma), and 25 µM pantothenate (Sigma) along with supplements above: EGF, hydrocortisone, glutamine, HI-FBS. Media was buffered with sodium hydroxide to pH 7.95. Designated plates were placed in hypoxia chamber at 1% oxygen. Cell number was measured at the indicated time points using sulforhodamine B (Sigma) staining, as previously described [42]. Absorbance at 510 nm was read on an Epoch plate reader (Bio-Tek). Relative growth was determined as (treatment absorbance / control absorbance).

## Statistics and reproducibility

Sample sizes, reproducibility, and statistical tests for each figure are denoted in the figure legends. All replicates are biological unless otherwise noted. Principal component analysis and nonhierarchical clustering were performed using Metaboanalyst 4.0 [40]. Graphics and accompanying statistical tests were designed on Prism 7. All error bars represent SEM, and significance between conditions is denoted as *, $p<0.05$; **, $p<0.01$; ***, $p<0.001$; and ****, $p<0.0001$.

# Results

## Hypoxia alters the polar metabolite profile of endothelial cells

In order to assess the effects of hypoxia on endothelial cells, we used HMEC-1, human dermal microvascular endothelial cells, which most closely mimic endothelial cells found in the tumor vasculature [43]. Exposure of HMEC-1 cells to hypoxia led to the expected increase of hypoxia-induced factor 1a protein (HIF1α). HIF1α levels peaked between 4 and 8 hrs of hypoxia exposure (Fig 1A), consistent with previous observations in microvascular endothelial cells [44]. The mRNA expression of HIF1α target genes, vascular endothelial growth factor (VEGF), solute carrier family 2 member 1 (SLC2A1, which encodes glucose transporter 1

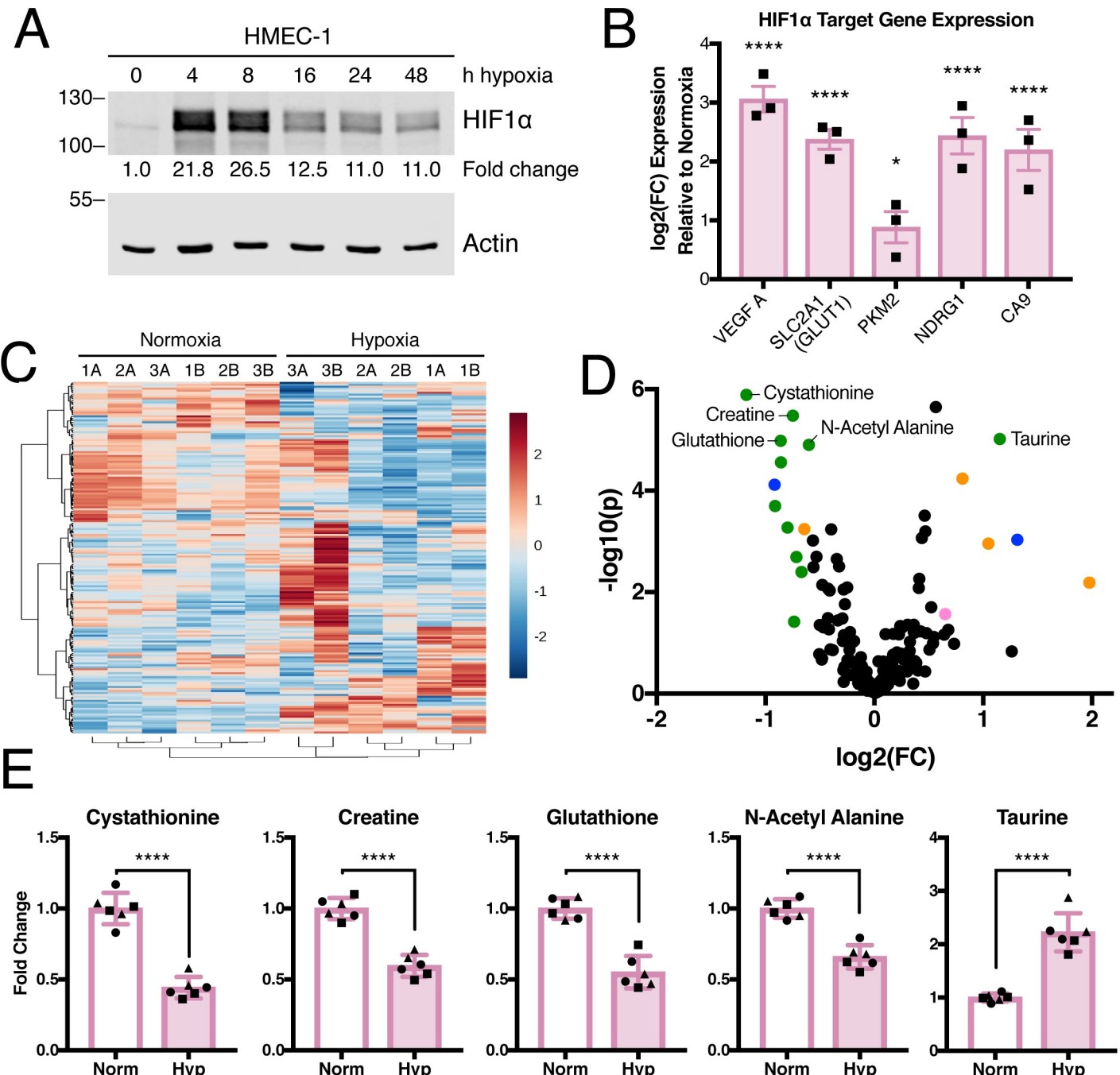

**Fig 1. Hypoxia alters the metabolite profile of microvascular cells.** (A) Lysates were prepared from HMEC-1 cells exposed to a time course of hypoxia and immunoblotted for HIF1α expression. Relative HIF1α expression was calculated using LI-COR Image Studio and is expressed as fold change relative to time = 0 sample. Representative of N = 1. (B) mRNA abundance of HIF1α target genes in HMEC-1 cells at 48 hrs of normoxia or hypoxia exposure was measured by quantitative real-time polymerase chain reaction (qRT-PCR) and is expressed as log(2) fold change relative to HMEC-1 cells cultured in normoxia, N = 3 biological replicates. Significance by two-way ANOVA. (C) Fold change metabolite abundance over normoxia control for 178 metabolites detected in all samples as measured by LC-MS/MS in HMEC-1 cells grown in hypoxia for 48 hrs. Number indicates biological replicate, letter represents technical replicate. (D) Effect size and significance by t-test of metabolite changes measured in C. Colored circles indicate p < 0.05 and fold change > 1.5. Top five most significant metabolites over 1.5-fold change are labeled. Colors indicate pathways: green, amino acid metabolism; orange, nucleotide metabolism; pink, sugar and energy metabolism; blue, other. (E) Fold change relative to normoxia of top five most significantly altered metabolites with over 1.5-fold change after 48 hrs in hypoxia. Different symbols represent biological replicates. Significance by unpaired t-test. All error bars represent SEM. *, p < 0.05; ****, p < 0.0001.

(GLUT1)), pyruvate kinase 2 (PKM2), N-myc downstream regulated 1 (NDRG1), and carbonic anhydrase 9 (CA9), also increased in response to hypoxia (Fig 1B) [44,45]. As expected, the proliferation of HMEC-1 cells was significantly reduced in hypoxia compared to cells grown under normoxic conditions (S1A Fig).

To examine the metabolic profile of HMEC-1 cells exposed to short-term hypoxia, we used targeted liquid chromatography tandem mass spectrometry (LC/MS-MS) to measure the levels of 265 polar metabolites in cells cultured in hypoxia for 48 hrs; 178 metabolites were detected in all samples (S1 Table). Hypoxia induced a consistent decrease in a subset of metabolites (Fig 1C). The fold change and corresponding significance was measured for each metabolite and revealed significant changes in amino acid-related metabolites (Fig 1D). The five most significantly altered metabolites by at least 1.5-fold are components of the intersecting pathways of cysteine-methionine (cystathionine), serine-threonine (creatine), glutathione, and taurine metabolism as well as N-acetyl-β-alanine, which comprises β-alanine metabolism (Fig 1E).

## Long-term hypoxia alters the polar metabolite profile of endothelial cells

To explore the long-term effects of hypoxia on endothelial cell metabolism, we measured polar metabolites in HMEC-1 cells cultured for 3 weeks in hypoxia (Fig 2A and S1 Table). While some pathways were affected by both 48 hrs and 3 weeks of hypoxia exposure, the metabolic profiles in short- and long-term hypoxia were distinct (S1C and S1D Fig). Both short- and long-term hypoxia exposure led to a statistically significant change in metabolites from amino acid and nucleotide metabolic pathways (Figs 2B and 1D and S1E Fig). Several metabolites involved in sugar and energy metabolism were exclusively altered following 3 weeks of hypoxia exposure (Fig 2B). Notably, four of the six sugar pathway metabolites significantly altered by long-term hypoxia exposure decreased and are involved in glycolysis or a connected pathway (Fig 2B and S2A and S2B Fig). Other glycolytic metabolites were also significantly altered in response to hypoxia (S2C Fig). The cofactors, thiamine pyrophosphate (TPP) and nicotinamide adenine dinucleotide (NADH), were also decreased significantly after 3 weeks of hypoxia (Fig 2B and S2D and S2E Fig); both are necessary for the activity of pyruvate dehydrogenase, which synthesizes acetyl CoA, a critical component of the TCA cycle and many other metabolic reactions.

In both short- and long-term hypoxia, the specific pathways that were most significantly altered and were predicted to be most impacted by the changes in the pathway analysis [40] were related to amino acid, nucleotide, and sugar metabolism (Fig 2C and S2 Table) [40]. These include taurine, cysteine-methionine, and glycine-serine-threonine metabolism. Alanine-aspartate-glutamate, arginine, nicotinate, ascorbate, and purine metabolism were also predicted to be impacted by hypoxia-induced changes in metabolite abundance. Many of the top 25 most significantly altered metabolites from long- and short-term hypoxia exposure are components of these affected pathways (Fig 2D).

## Hypoxia decreases the abundance of many cysteine-derived metabolites

Three of the top ten metabolic pathways in HMEC-1 cells most significantly altered by hypoxia exposure are interconnected: taurine, cysteine-methionine, and glycine-serine-threonine metabolism (Fig 3A and S2 Table). These pathways are also related to glutathione metabolism (Fig 3A), which was also altered by hypoxia, but not significantly (S2 Table). The most significantly decreased metabolite in hypoxia is cystathionine, which is shared amongst all four pathways (S1E Fig). Many metabolites in these pathways were decreased under hypoxia (Fig 3B). Significant changes in twelve metabolites from these pathways were observed after 48 hrs or 3 weeks in hypoxia (Fig 3C–3F). Levels of several metabolites were decreased at both time

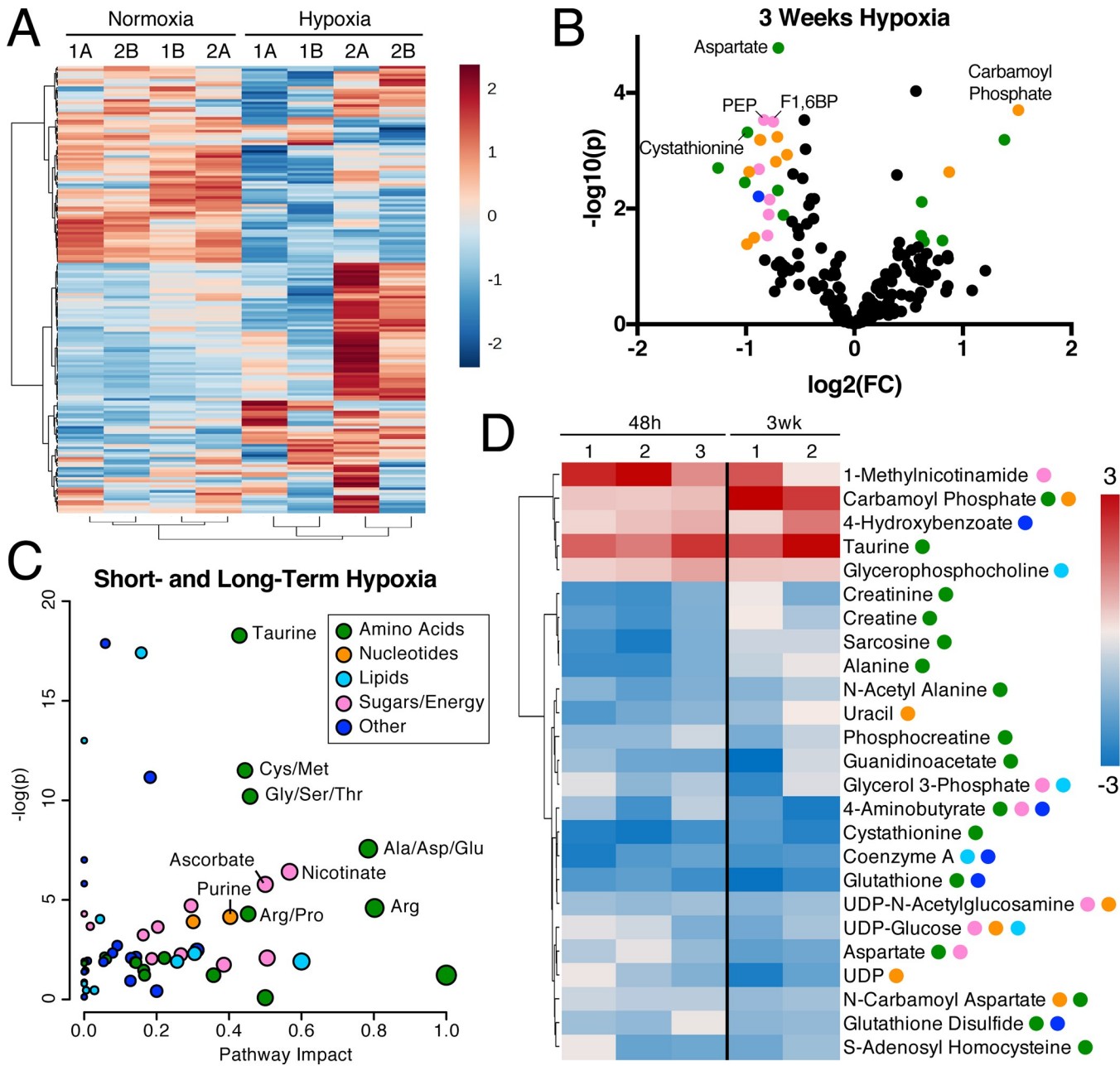

**Fig 2. Long-term hypoxia alters the metabolite profile of microvascular cells.** (A) Fold change metabolite abundance over normoxia control for 175 metabolites detected in all samples as measured by LC-MS/MS in HMEC-1 cells grown in hypoxia for 3 weeks. Number indicates biological replicate, letter represents technical replicate. (B) Effect size and significance by t-test of metabolite changes measured in A. Colored circles indicate p < 0.05 and fold change > 1.5. Top five most significant metabolites over 1.5-fold change are labeled. Colors indicate pathways: green, amino acid metabolism; orange, nucleotide metabolism; pink, sugar and energy metabolism; blue, other. (C) Metaboanalyst 4.0 Pathway Analysis comparing effects of hypoxia at 48 hrs and 3 weeks to normoxia. Metabolites measured at only one time point were excluded for a total of 150 metabolites detected in all samples. (D) Top 25 most significantly altered metabolites in hypoxia at 48 hrs and 3 weeks compared to normoxia controls. Abbreviations: Ala, alanine; Arg, arginine; Asp, aspartate; Cys, cysteine; F1,6BP, fructose-1,6-bisphosphate; Glu, glutamate; Gly, glycine; Met, methionine; PEP, phophoenolpyruvate; Pro, proline; Ser, serine; Thr, threonine; UDP, uridine diphosphate.

points, including oxidized and reduced glutathione, S-adenosyl-homocysteine, cystathionine, and guanidinoacetate (Fig 3C, 3D and 3F). Taurine was the only related metabolite that increased in both short- and long-term hypoxia (Fig 3E).

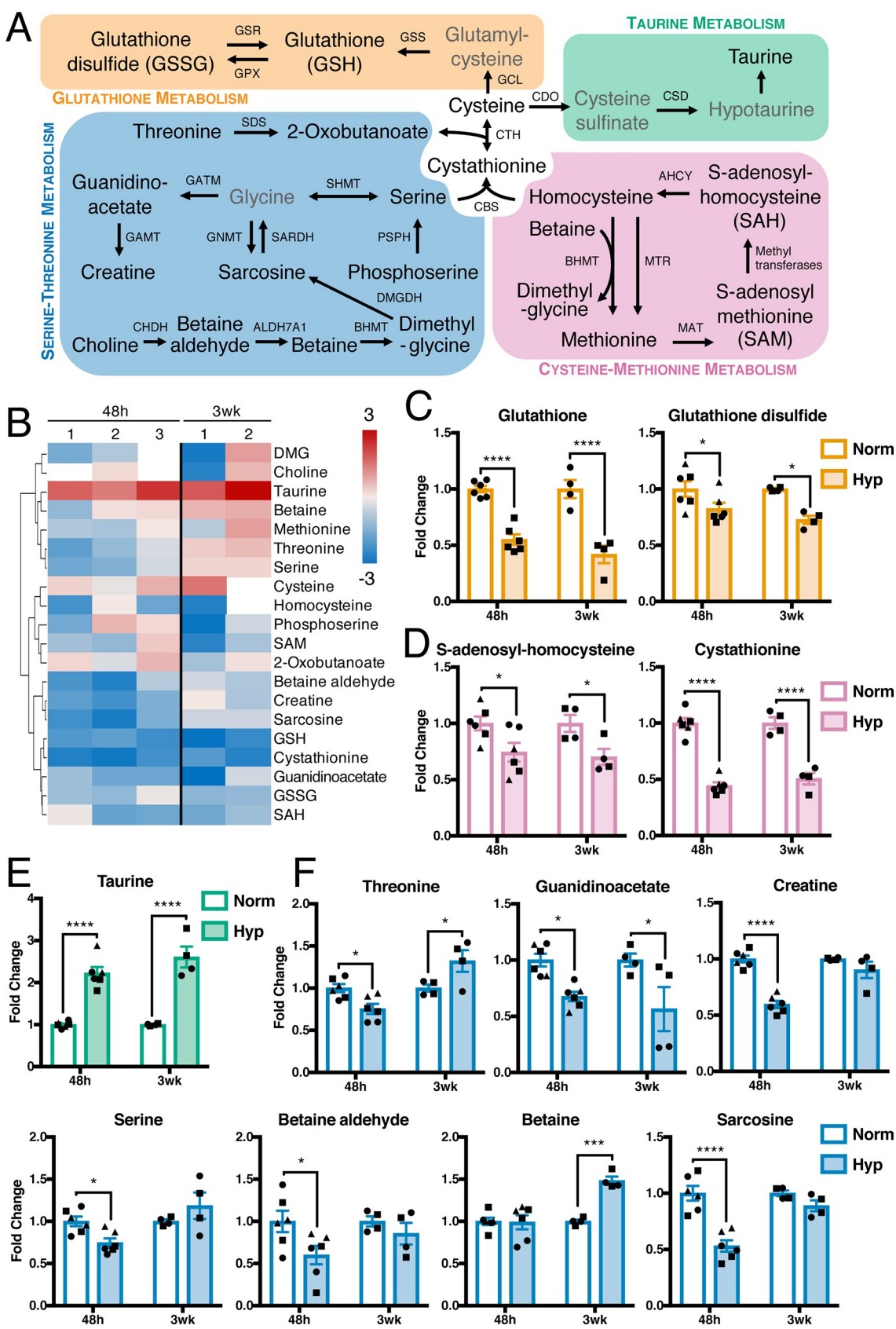

**Fig 3. Interconnected pathways of serine and cysteine metabolism are altered by hypoxia.** (A) Metabolites and enzymes in serine-threonine, glutathione, taurine, and cysteine-methionine metabolism. Metabolites in gray not measured or detected. (B) Fold change metabolite abundance over normoxia control for metabolites from pathway in A detected in more than one sample. White space indicates metabolite not detected. (C-F) Significantly altered metabolites from B. Significance by two-way ANOVA. All error bars represent SEM, different symbols represent biological replicates. *, $p < 0.05$; ***, $p < 0.001$; ****, $p < 0.0001$. Enzyme abbreviations: AHCY, Adenosylhomocysteinase; ALDH7A1, aldehyde dehydrogenase 7 family member A1; BHMT, betaine—homocysteine S-methyltransferase; CBS, cystathionine beta-synthase; CDO, cysteine dioxygenase; CHDH, choline dehydrogenase; CSD, cysteine desulfurase; CTH, cystathionine gamma-lyase; DMGDH, dimethylglycine dehydrogenase; GAMT, guanidinoacetate N-methyltransferase; GATM, glycine amidinotransferase; GCL, glutamate cysteine ligase; GNMT, glycine N-methyltransferase; GPX, glutathione peroxidase; GSR, glutathione-disulfide reductase; GSS, glutathione synthetase; MAT, methionine adenosyltransferase; MTR, methionine synthase; PSPH, phosphoserine phosphatase; SARDH, sarcosine dehydrogenase; SDS, serine dehydratase; SHMT, serine hydroxymethyltransferase. Metabolite abbreviations: DMG, dimethylglycine; GSH, glutathione; GSSG, glutathione disulfide; SAH, S-adenosylhoomocysteine; SAM, S-adenosylmethionine.

Due to the numerous alterations in these pathways, we hypothesized that hypoxia affects key nodes of this network. Cysteine and cystathionine connect these four pathways (Fig 3A). While consistent changes in cysteine levels after hypoxia exposure were not observed, many cysteine-derived metabolites were decreased (Fig 3B). Cysteine can be synthesized intracellularly through the transsulfuration pathway or transported into cells by xCT in its oxidized form, cystine [46]. We confirmed that HMEC-1 cells express the cystine/glutamate antiporter, xCT, and observed a decrease in xCT levels after hypoxia exposure (S3A Fig). However, overexpression of xCT failed to rescue growth in HMEC-1 cells in hypoxia and instead reduced growth under all conditions (S3B and S3C Fig), likely due to its function as a glutamate antiporter [47]. Moreover, transcript levels of cysteine-metabolizing enzymes were not affected by hypoxia exposure (S3D Fig).

Glutathione and taurine both exert protective effects from oxidative stress and are synthesized from cysteine (Fig 3A) [46]. However, in HMEC-1 cells exposed to hypoxia, glutathione levels decreased and taurine increased (Fig 3C and 3E). We hypothesized that in hypoxia, HMEC-1 cells preferentially synthesize taurine at the expense of glutathione. However, $^{13}C_3$$^{15}N$-cysteine tracing did not reveal decreased labeling of glutathione under hypoxia, indicating the decreased abundance is not due to deficient *de novo* synthesis (S3E and S3F Fig). We also did not observe significant labeling of hypotaurine or taurine in the last 16 hrs of short-term hypoxia exposure, suggesting that sustained elevation of synthesis does not contribute to their elevation in hypoxia (S3G Fig). Because the decrease in glutathione could reduce the ability of HMEC-1 cells to buffer the toxic effects of reactive oxygen species (ROS) in hypoxia, we also tested the ability of glutathione and glutathione precursors to rescue the growth defect of HMEC-1 cells in hypoxia. However, addition of the cell permeable, oxidized form of homocysteine (homocystine), or the cell-permeable forms of cysteine (N-acetyl cysteine) or glutathione (glutathione ethyl ester) did not rescue proliferation of HMEC-1 cells under hypoxia (S3H–S3J Fig).

## Hypoxia decreases aspartate availability and alters levels of related metabolites

The metabolism of the amino acids, alanine, aspartate, and glutamate, lies at the center of many critical metabolic pathways: the TCA cycle, glycine-serine-threonine metabolism, pyrimidine metabolism, and arginine metabolism (Fig 4A). The alanine-aspartate-glutamate pathway was one of the most significantly altered and impacted pathways in short- and long-term hypoxia (S2 Table). Several metabolites in this pathway were increased following hypoxia exposure and a few decreased, some significantly (Fig 4B and 4C). Aspartate was the only metabolite that consistently and significantly decreased after both short- and long-term hypoxia. We hypothesized that this may be due to alterations in metabolic reactions that consume

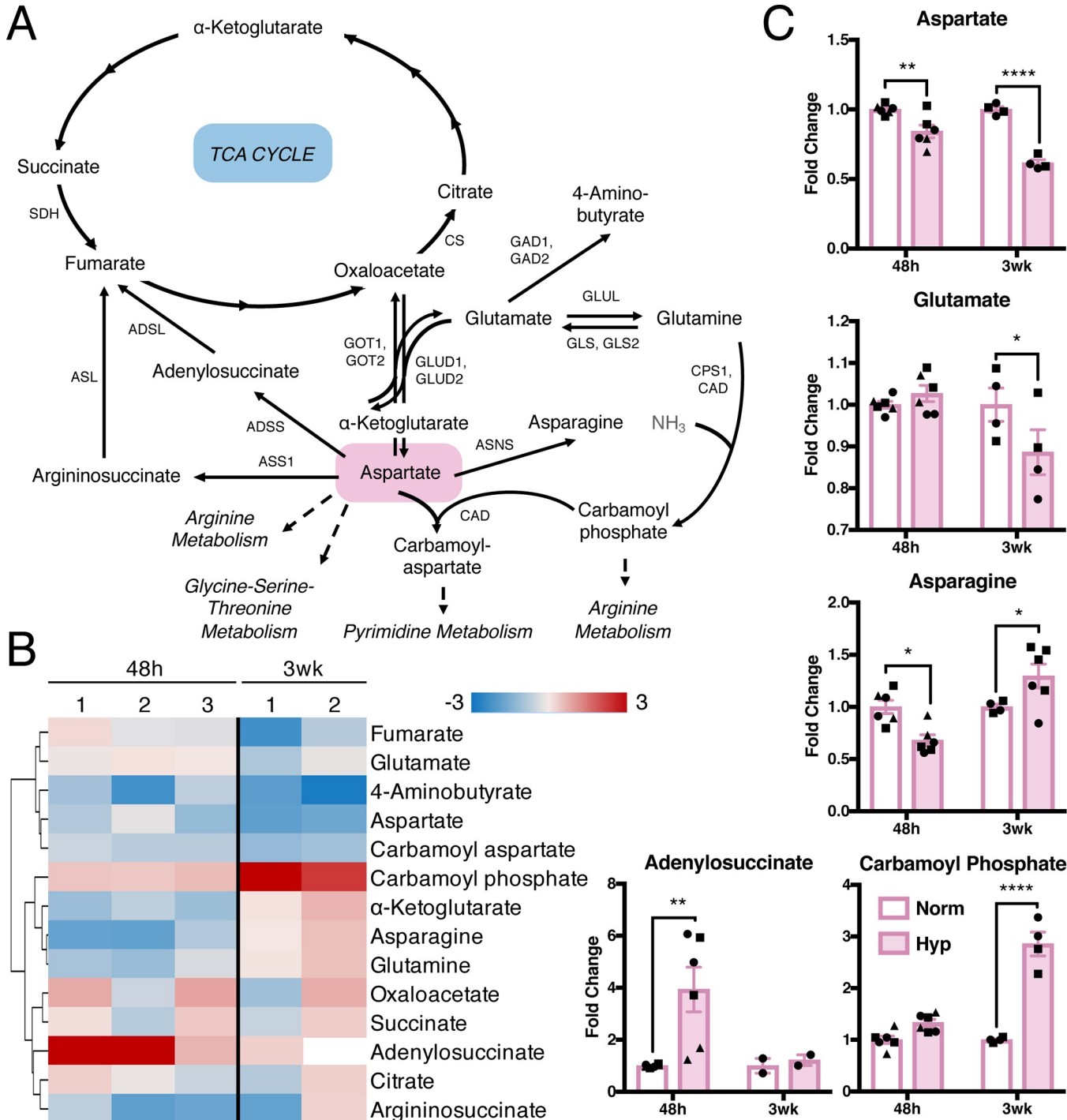

**Fig 4. Aspartate and its metabolites are decreased in hypoxia.** (A) Metabolites and enzymes in aspartate metabolism. Metabolites in gray not measured or detected. (B) Fold change metabolite abundance over normoxia control for metabolites in pathway A detected in more than one sample. White space indicates metabolite not detected. (C) Significantly altered metabolites from B. Significance by two-way ANOVA. All error bars represent SEM, different symbols represent biological replicates. *, $p < 0.05$; **, $p < 0.01$; ****, $p < 0.0001$. Abbreviations: ADSL, adenylosuccinate lyase; ADSS, adenylosuccinate synthase; ASL, argininosuccinate lyase; ASNS, asparagine synthetase; ASS1, argininosuccinate synthase; CAD, carbamoyl-phosphate synthetase 2, aspartate transcarbamylase, and dihydroorotase; CPS1, carbamoyl phosphate synthetase; CS, citrate synthase; GAD1/2, glutamate decarboxylase 1/2; GLUD1/2, glutamate dehydrogenase 1/2; GLS/GLS2, glutaminase 1/2; GLUL, glutamine synthetase; GOT1/2, aspartate aminotransferase, cytoplasmic/mitochondrial; SDH, succinate dehydrogenase.

or produce aspartate, although hypoxia did not alter transcript levels of enzymes that participate in aspartate metabolism (S4A Fig).

Several recent studies have demonstrated an important role for aspartate in cancer cell proliferation under hypoxia, electron transport chain (ETC)-deficient or inhibited conditions, and in glutamine-limited settings [48–53]. Treating HMEC-1 cells with high levels of exogenous aspartate did not affect their growth in hypoxia or normoxia (S4B Fig). Since many cells cannot take up exogenous aspartate due to no or low expression of the aspartate transporter, SLC1A3 [50], we also expressed SLC1A3 prior to aspartate supplementation to levels comparable to MDA-MB-468 breast cancer cells (S4C Fig) [54]. However, addition of exogenous aspartate failed to rescue growth of SLC1A3-expressing HMEC-1 cells in hypoxia (S4D Fig). Since previous studies in tumor cells indicated that aspartate is needed for proliferative growth by regenerating electron acceptors [49], we also attempted to rescue growth of HMEC-1 cells in hypoxia by supplementation of the electron acceptors, α-ketoglutarate, α-ketobutyrate, and pyruvate; however, this did not rescue the growth defect in HMEC-1 cells under hypoxia or normoxia (S4E–S4G Fig).

## Nucleotides are depleted in hypoxia, while precursors accumulate

Purine metabolism represents one branch of nucleotide metabolism and is dependent on aspartate availability. Purines can be classified as adenylate and guanylate nucleotides (Fig 5A). Many of the metabolites in adenylate and guanylate metabolism were decreased after long-term hypoxia (Fig 5B–5E). However, none of the metabolites in purine metabolism significantly and consistently changed in both short- and long-term hypoxia.

Pyrimidine metabolism is the other branch of nucleotide metabolism, and consists of three sub-pathways: *de novo* synthesis, cytidine metabolism, and thymidine metabolism (Fig 6A). *De novo* pyrimidine synthesis is also connected to aspartate and arginine metabolism, where we observed significant alterations (Fig 4 and S5 Fig). Many pyrimidine-related metabolites decreased following hypoxia exposure, particularly after 3 weeks (Fig 6B). Only carbamoyl phosphate increased subsequent to long-term hypoxia, and other upstream metabolites, including orotate and phosphoribosyl pyrophosphate, also increased in either short- or long-term hypoxia (Fig 6C). Most metabolites further downstream in the pathway were decreased with the exception of thymine, which was increased after short-term hypoxia (Fig 6D–6F).

## Discussion

Understanding endothelial cell metabolism in the context of the tumor vasculature is critical to developing efficient and effective anti-cancer therapies. While many AATs are approved by regulatory agencies for various cancers [14], resistance frequently develops due to inefficient drug delivery and tumor adaption to a hypoxic environment [1,3,6–9]. Normalizing the tumor vasculature is key to allowing efficient therapy delivery [1,3,23]. We approached this therapeutic challenge from a metabolic perspective with the goal of identifying metabolic alterations that inform our understanding of the state of endothelial cells in a tumor. We focused on microvasculature endothelial cell metabolism in hypoxia to model tumor endothelial cells in a hypoxic microenvironment.

Our results revealed that the metabolism of amino acids and nucleotides were altered after both short- and long-term hypoxia. The cysteine-derived antioxidant, glutathione, decreased in hypoxia (Fig 3C) while taurine, another metabolite synthesized from cysteine that counteracts oxidative stress increased (Fig 3E). This is indicative of a potential shift that may enable endothelial cells to cope with oxidative stress under hypoxic conditions. This is consistent with previous observations in which hypoxia increases ROS in a mitochondria-dependent manner

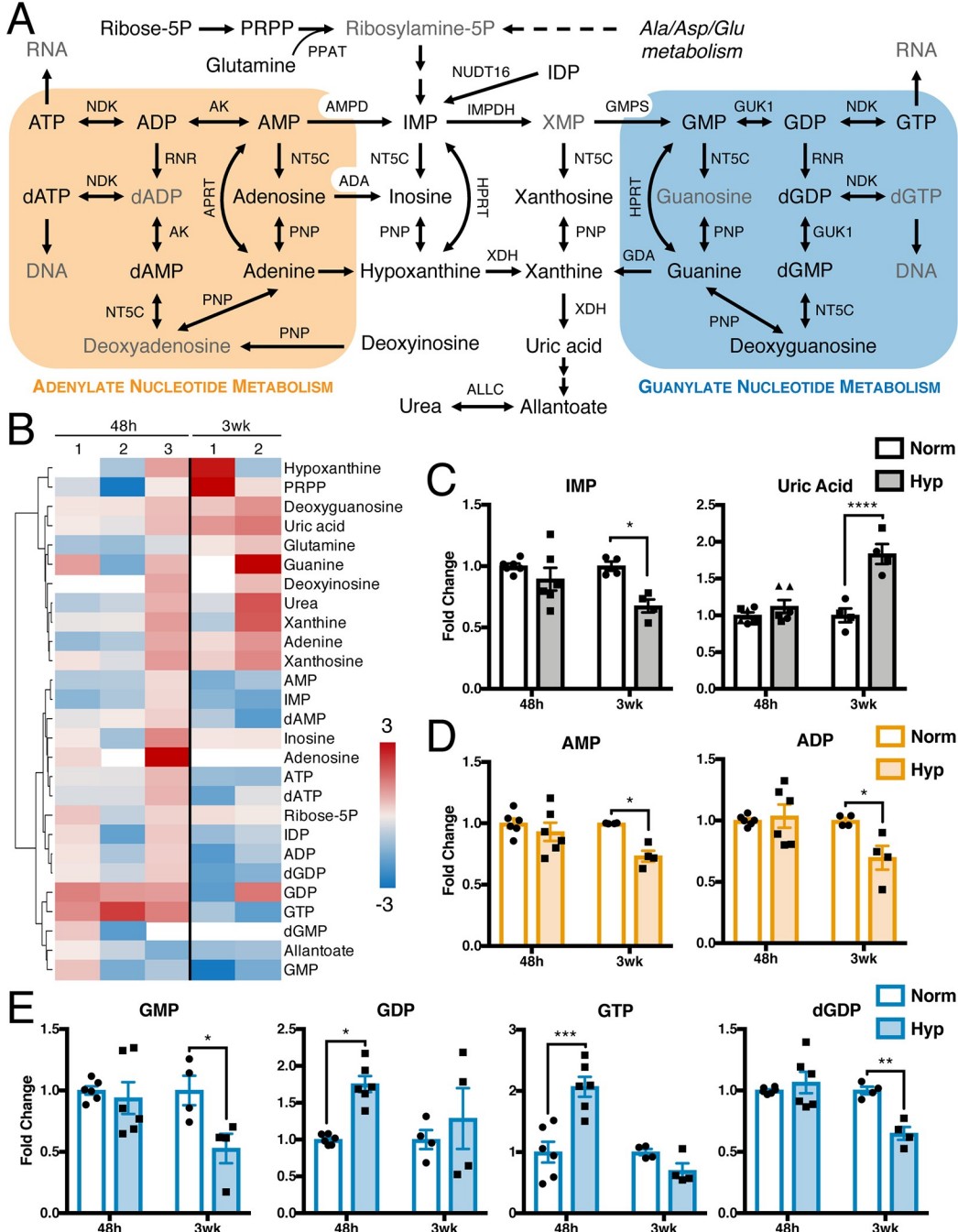

**Fig 5. Purine nucleotides are decreased in long-term hypoxia.** (A) Metabolites and enzymes in purine metabolism. Metabolites in gray not measured or detected. (B) Fold change metabolite abundance over normoxia control for metabolites from pathway in A detected in more than one sample. White space indicates metabolite not detected. (C-E) Significantly altered metabolites from B. Significance by two-way ANOVA. All error bars represent SEM, different symbols represent biological replicates. *, $p < 0.05$; **, $p < 0.01$; ***, $p < 0.001$; ****, $p < 0.0001$. Enzyme abbreviations: ADA, adenosine deaminase; AK, adenosine kinase; ALLC, allantoicase; AMPD, AMP deaminase; GDA, guanine deaminase; GMPS, GMP synthase; GUK1, guanylate kinase; HPRT, hypoxanthine-guanine phosphoribosyltransferase; IMPDH, IMP dehydrogenase; NDK, nucleoside diphosphate kinase; NT5C, 5', 3'-nucleotidase, cytosolic; NUDT16, IDP phosphatase; PNP, purine nucleoside phosphorylase; PPAT, phosphoribosyl pyrophosphate amidotransferase; RNR, ribonucleoside-diphosphate reductase; XDH, xanthine dehydrogenase. Metabolite abbreviations: ADP, adenosine diphosphate; AMP, adenosine monophosphate; ATP, adenosine triphosphate; dADP, deoxyadenosine diphosphate; dAMP, deoxyadenosine monophosphate; dATP, deoxyadenosine triphosphate; dGDP, deoxyguanosine diphosphate; dGMP, deoxyguanosine monophosphate; dGTP, deoxyguanosine triphosphate; GDP, guanosine diphosphate; GMP, guanosine monophosphate;

GTP, guanosine triphosphate; IDP, inosine diphosphate; IMP, inosine monophosphate; PRPP, phosphoribosyl pyrophosphate; Ribose-5P, ribose 5-phosphate; XMP, xanthosine monophosphate.

[55], including in endothelial cells [56,57], leading to a decrease in glutathione and a corresponding reduction of cystine uptake [58]. Taurine levels are known to increase during inflammation or oxidative stress [59] and in other cell types exposed to hypoxia [60,61]. Since addition of glutathione/taurine precursors or glutathione did not affect the proliferation of HMEC-1 cells in hypoxia, and hypoxia did not alter transcript levels of cysteine-metabolizing enzymes or cysteine labeling of glutathione and taurine, the decreased glutathione levels could be due to efflux and/or degradation. Multidrug resistance protein-1 (MRP1), an ABC transporter, has a high affinity for the oxidized form of glutathione, GSSG [62], and under oxidative stress several cell types enhance GSSG export via MRP [62]. This has been observed in endothelial cells along with a concomitant decrease in intracellular reduced glutathione [63].

The metabolic pathways central to the amino acid aspartate were also altered in HMEC-1 cells in hypoxia. Aspartate metabolism is a key node that connects the TCA cycle, arginine, glycine-serine-threonine and pyrimidine metabolism. Changes in oxidative metabolism in hypoxic endothelial cells have been reported [57] and may explain, at least in part, our observations of altered antioxidant and aspartate levels. Decreased mitochondrial function of endothelial cells in hypoxia [57] could also account for our observation of decreased aspartate, presumably due to lack of electron acceptors necessary for both mitochondrial function and aspartate synthesis. The observed decrease in aspartate could in turn lead to observed decreased levels of carbamoyl aspartate and pyrimidine nucleotides. The precise mechanism by which aspartate is decreased in hypoxia remains to be determined, although a block in synthesis or accelerated consumption are likely to contribute.

Decreased aspartate levels have been observed in cancer cells in hypoxia or ETC-deficient or inhibited conditions [48–51,53]. Studies have shown that decreased aspartate is due to a deficiency of electron acceptors necessary for aspartate production by the TCA cycle in ETC-deficient cancer cells [49]. However, in our study we found that neither aspartate nor electron acceptors could to rescue the growth defect of HMEC-1 cells in hypoxia, indicating that these are not the primary mechanisms for reduced cell proliferation (S4B and S4D–S4G Fig). In this context, ETC inhibition in HUVEC endothelial cells decreased aspartate levels with a concomitant inhibition of cell proliferation, but aspartate addition did not rescue proliferation [64]. Despite differences in endothelial cell types and perturbations, these findings suggest that aspartate has unique roles in endothelial cells experiencing oxidative stress, and which differ from the functional mechanisms that are found in tumor cells.

Studies in cancer cells that reported decreased aspartate under oxidative stress also proposed reduction of aspartate levels as a therapeutic approach in cancer [50–52]. However, our observations of decreased aspartate in endothelial cells in hypoxia indicate that this would likely be detrimental to the tumor vasculature, enhancing the aspartate-limiting effect of hypoxia exposure on vessel cells, which in turn would blunt the therapeutic benefit to tumors through compromising the vasculature and leading to poor drug delivery.

Our study highlights the profile of endothelial cell metabolic pathways and further highlights distinctions between tumor cell and endothelial cell metabolism under hypoxia. We observed decreased glycolytic intermediates in HMEC-1 cells after long-term hypoxia, while other studies in distinct types of endothelial cells in hypoxia or of tumor endothelial cells *in vivo* observed increased glycolysis [34,65–67]. By contrast, our data are consistent with previous observations that detected a decrease in glycolytic metabolite intermediates observed in a mouse model with endothelial cell-specific ETC deficiency [64]. These results highlight the

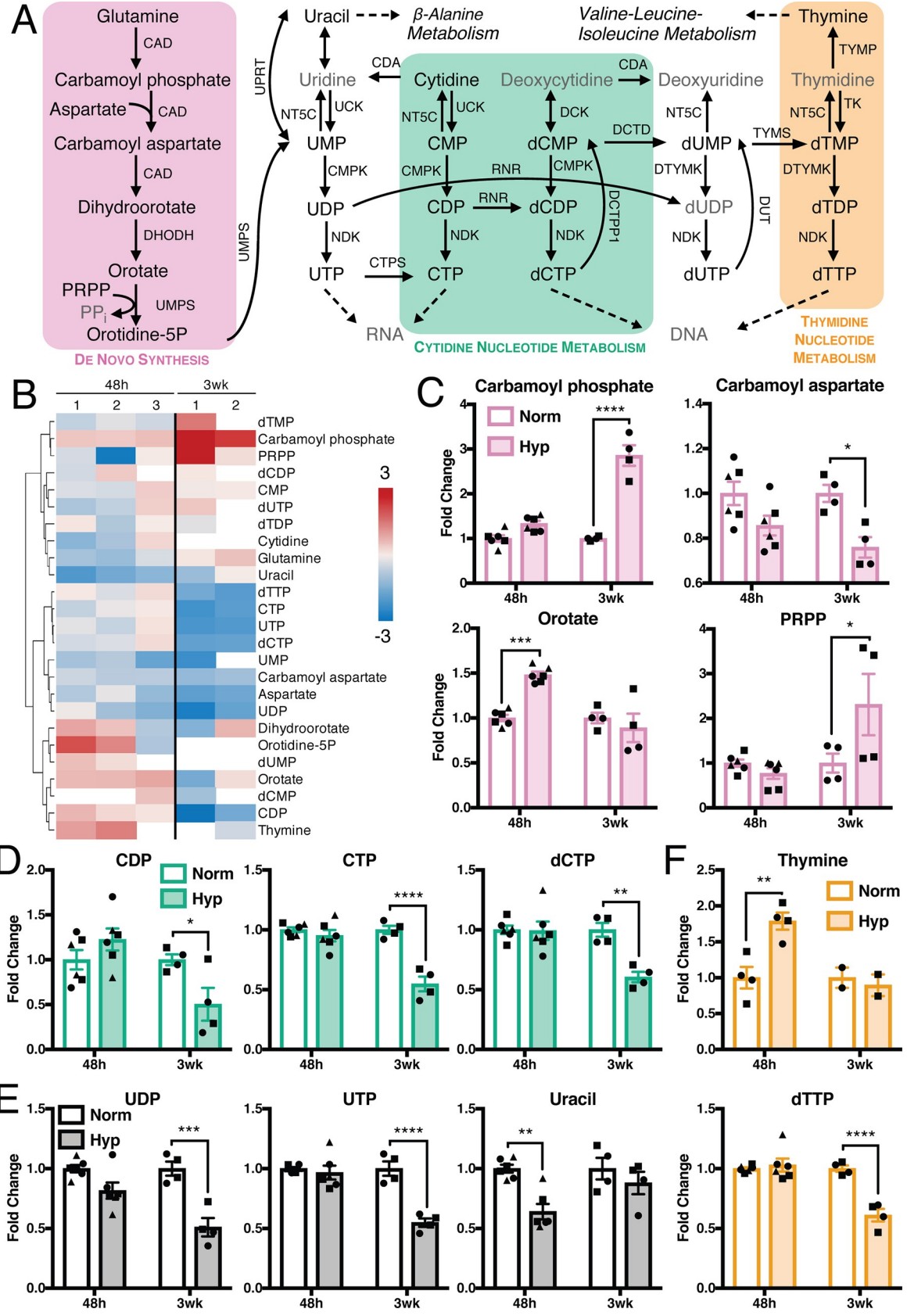

**Fig 6. Pyrimidine nucleotides are decreased in hypoxia, while precursors accumulate.** (A) Metabolites and enzymes in pyrimidine metabolism. Metabolites in gray not measured or detected. (B) Fold change metabolite abundance over normoxia control for metabolites in pathway A detected in more than one sample. White space indicates metabolite not detected. (C-E) Significantly altered metabolites from B. Significance by two-way ANOVA. All error bars represent SEM, different symbols represent biological replicates. *, $p < 0.05$; **, $p < 0.01$; ***, $p < 0.001$; ****, $p < 0.0001$. Enzyme abbreviations: CAD, carbamoyl-phosphate synthetase 2, aspartate transcarbamylase, and dihydroorotase; CDA, cytidine deaminase; CMPK, UMP-CMP kinase; CTPS, CTP synthase; DCK, deoxycytidine kinase; DCTD, deoxycytidylate deaminase; DCTPP1, dCTP pyrophosphatase 1; DHODH, dihydroorotate dehydrogenase; DTYMK, thymidylate kinase; DUT, dUTP nucleotidohydrolase; NDK, nucleoside diphosphate kinase; NT5C, 5', 3'-nucleotidase, cytosolic; RNR, ribonucleoside-diphosphate reductase; TK, thymidine kinase; TYMP, thymidine phosphorylase; TYMS, thymidylate synthase; UCK, uridine-cytidine kinase; UMPS, UMP synthase; UPRT, uracil phosphoribosyltransferase. Metabolite abbreviations: CDP, cytidine diphosphate; CMP, cytidine monophosphate; CTP, cytidine triphosphate; dCDP, deoxycytidine diphosphate; dCMP, deoxycytidine monophosphate; dCTP, deoxycytidine triphosphate; dTDP, deoxythymidine diphosphate; dTMP, deoxythymidine monophosphate; dTTP, deoxythymidine triphosphate; dUDP, deoxyuridine diphosphate; dUMP, deoxyuridine monophosphate; dUTP, deoxyuridine triphosphate; orotidine-5P, orotidine 5-phosphate; PPi, pyrophosphate; PRPP, phosphoribosyl pyrophosphate; UDP, uridine diphosphate; UMP, uridine monophosphate; UTP, uridine triphosphate.

need for development of models that accurately represent tumor endothelial cells exposed to hypoxia in order to identify robust metabolic differences. Within a tumor, endothelial cells do not experience continual hypoxia, but rather alternating oxygen concentrations [32]. While our cells maintained in long-term hypoxia were exposed to intermittent normoxia for maintenance of the growth conditions, a more frequent and controlled oscillation would more accurately model physiological exposure to normoxia and hypoxia. Our analysis of metabolic profiling are limited by the polar metabolites detected on our targeted LC-MS/MS platform; further studies of other metabolites, particularly oxidation of fatty acids, would provide additional information that has been garnered in other studies [57]. Future studies on the metabolic reprogramming in co-cultures of endothelial cells and tumor cells under hypoxia are also likely to provide much needed information, as well as determining angiogenic-specific phenotypes such as tube formation and the integrity of VE-cadherin junctions. Other studies have shown that hypoxia-induced mitochondrial ROS mediates increased endothelial cell permeability [56], indicating that the gross morphology of the tumor vasculature could be affected by hypoxia.

These findings are of broad relevance to the vasculature in the context of other pathophysiologies, including cardiovascular disorders, ocular diseases, diabetes, and brain aneurysms, as well as to normal physiologies exposed to hypoxia [32,33,36,68]. Many of these disease states exhibit increased ROS similar to what is observed in a hypoxic tumor. Considering the many endothelial cell vascular diseases exhibit metabolic dysfunction, these findings may inform the development of more efficient and effective therapeutic strategies for managing a wide range of human diseases.

## Supporting information

**S1 Fig. Supporting information for polar metabolite profiling in hypoxia.** (A) HMEC-1 cells were grown in normoxia or hypoxia for six days. Proliferation of the cells was determined using sulforhodamine B staining at the indicated time points, N = 4 biological replicates. (B) mRNA abundance of HIF1α target genes from 3-week metabolomics-matched RNA samples were measured by qRT-PCR and is expressed as log(2) fold change relative to HMEC-1 cells cultured in normoxia, N = 2 biological replicates. (C) Fold change metabolite abundance over normoxia control for 150 metabolites detected in all samples as measured by LC-MS/MS of HMEC-1 cells grown in hypoxia for 48 hrs and 3 weeks. (D) Principal component analysis of data from C. (E) Effect size and significance by t-test of metabolite changes measured in C. Colored circles indicate p < 0.05 and fold change > 1.5. Colors indicate pathways: green, amino acid metabolism; orange, nucleotide metabolism; blue, other. All error bars represent

SEM. Significance by two-way ANOVA except E as described above. *, p < 0.05; ****, p < 0.0001.
(TIF)

**S2 Fig. Glycolytic intermediates are depleted in hypoxia.** (A) Metabolites and enzymes in glycolysis. Metabolites in gray not measured or detected. (B) Fold change metabolite abundance over normoxia control for metabolites in pathway A detected in more than one sample. White space indicates metabolite not detected. (C) Significantly altered metabolites from B. (D-E) Fold change abundance compared to normoxia of cofactors for glycolytic enzymes. All error bars represent SEM, different symbols represent biological replicates. Significance by two-way ANOVA except D by unpaired t-test. *, p < 0.05; **, p < 0.01; ***, p < 0.001. Enzyme abbreviations: ALDO, aldolase; ENO, enolase; FBP, fructose-1, 6-bisphosphatase; G6PC, glucose-6-phosphatase; GAPDH, glyceraldehyde 3-phosphate dehydrogenase; GOT, aspartate aminotransferase; GPI, glucose-6-phosphate isomerase; HK, hexokinase; LDH, lactate dehydrogenase; MDH, malate dehydrogenase; PC, pyruvate carboxylase; PCK1, phosphoenolpyruvate carboxykinase; PFK, phosphofructose kinase; PGAM, phosphoglycerate mutase; PGK, phosphoglycerate kinase; PK, pyruvate kinase. Metabolite abbreviations: CoA, coenzyme A; DHAP, dihydroxyacetone phosphate; NAD+, oxidized nicotinamide adenine dinucleotide; NADH, nicotinamide adenine dinucleotide; P, phosphate; $P_2$, bisphosphate; PEP, phosphoenolpyruvate; TPP, thiamine pyrophosphate.
(TIF)

**S3 Fig. Biological characterization of cysteine metabolism in hypoxia.** (A) HMEC-1 cell lysates were prepared from all 48-hr and one 3-week metabolomics-matched protein samples and immunoblotted for xCT expression. Relative xCT expression in hypoxia was calculated using Bio-Rad Image Lab and is expressed as fold change relative to paired normoxia sample. (B) Lysates were prepared from naïve HMEC-1 cells or HMEC-1 cells overexpressing xCT and immunoblotted for xCT expression. (C) Naïve HMEC-1 cells or HMEC-1 cells overexpressing xCT were grown in normoxia or hypoxia for six days, and the proliferation of the cells was determined using SRB staining, N = 3 technical replicates. (D) mRNA abundance of cysteine metabolizing enzymes in HMEC-1 cells after 48 hrs of normoxia or hypoxia exposure was measured by qRT-PCR and is expressed as log(2) fold change relative to HMEC-1 cells cultured in normoxia, N = 3 biological replicates. (E-G) Mass isotopomer analysis by LC-MS/MS of (E) cysteine, (F) reduced and oxidized glutathione, and (G) hypotaurine and taurine in HMEC-1 cells cultured in normoxia or hypoxia for 48 hrs, the last 16 hrs of which in medium containing 165 μM U-$^{13}C_3$$^{15}N$-cysteine, N = 2 technical replicates. (H-J) HMEC-1 cells were grown in normoxia or hypoxia for six days in the presence or absence of (H) homocystine (Hcy), (I) N-acetyl cysteine (NAC), or (J) glutathione ethyl ester (GEE). Proliferation of the cells was determined using SRB staining, N = 3 technical replicates. All error bars represent SEM. Significance by two-way ANOVA. *, p < 0.05; **, p < 0.01; ***, p < 0.001; ****, p < 0.0001.
(TIF)

**S4 Fig. Aspartate and electron acceptors do not rescue growth defects in hypoxia.** (A) mRNA abundance of aspartate metabolism-associated enzymes in HMEC-1 cells after 48 hrs of normoxia or hypoxia exposure was measured by qRT-PCR and is expressed as log(2) fold change relative to HMEC-1 cells cultured in normoxia, N = 3 biological replicates. (B) HMEC-1 cells were grown in normoxia or hypoxia for six days in the presence or absence of 20 mM aspartate. Proliferation of the cells was determined using SRB staining, N = 2 biological replicates. (C) mRNA abundance of *SLC1A3* in HMEC-1 cells overexpressing SLC1A3 or naïve

MDA-MB-468 breast cancer cells was measured by qRT-PCR and is expressed as log(2) fold change relative to naïve HMEC-1 cells, N = 3 technical replicates. (D) HMEC-1 cells expressing an empty vector or SLC1A3 were grown in normoxia or hypoxia for six days in the presence or absence of 150 μM aspartate in normoxia or hypoxia. Proliferation of the cells was determined using SRB staining, N = 3 technical replicates. HMEC-1 cells were grown in normoxia or hypoxia for six days in the presence or absence of (E) 1 mM dimethyl α-ketoglutarate (AKG), (F) 1 mM α-ketobutyrate (AKB), (G) 2 mM pyruvate. Proliferation of the cells was determined using SRB staining, N = 3 technical replicates. All error bars represent SEM. Significance by two-way ANOVA. *, $p < 0.05$; ***, $p < 0.001$; ****, $p < 0.0001$.
(TIF)

**S5 Fig. Arginine metabolism is altered in hypoxia.** (A) Metabolites and enzymes in arginine metabolism. Metabolites in gray not measured or detected. (B) Fold change metabolite abundance over normoxia control for metabolites in pathway A detected in more than one sample. (C) Significantly altered metabolite from B not shown in prior figures. Significance by two-way ANOVA. All error bars represent SEM, different symbols represent biological replicates. *, $p < 0.05$. Abbreviations: ARG2, arginase 2; ASL, argininosuccinate lyase; ASS1: argininosuccinate synthase; CPS1, carbamoyl phosphate synthetase; NOS1/2/3, nitric oxide synthase 1/2/3; OTC, ornithine transcarbamylase.
(TIF)

**S1 Raw images. Full immunoblot images.**
(PDF)

**S1 Table. Metabolite abundance in short- and long-term hypoxia.**
(XLSX)

**S2 Table. Analysis of metabolic pathways altered by hypoxia.**
(XLSX)

**S3 Table. RT-PCR primers.**
(XLSX)

## Acknowledgments

We thank members of the Toker Lab and C. Dibble (BIDMC) for helpful discussions; J. Asara (BIDMC) and M. Yuan (BIDMC) for assistance with mass spectrometry.

## Author Contributions

**Conceptualization:** Emily B. Cohen, Renee C. Geck, Alex Toker.

**Data curation:** Emily B. Cohen, Renee C. Geck.

**Formal analysis:** Emily B. Cohen, Renee C. Geck.

**Funding acquisition:** Emily B. Cohen, Renee C. Geck.

**Investigation:** Emily B. Cohen, Renee C. Geck.

**Methodology:** Emily B. Cohen, Renee C. Geck, Alex Toker.

**Project administration:** Emily B. Cohen, Alex Toker.

**Resources:** Alex Toker.

**Supervision:** Alex Toker.

**Validation:** Emily B. Cohen, Renee C. Geck, Alex Toker.

**Visualization:** Emily B. Cohen, Renee C. Geck.

**Writing – original draft:** Emily B. Cohen.

**Writing – review & editing:** Emily B. Cohen, Renee C. Geck, Alex Toker.

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
