## [Decision Letter · Decision Letter 0]

5 May 2020

PONE-D-20-09862

Metabolic pathway alterations in microvascular endothelial cells in response to hypoxia

PLOS ONE

Dear Dr. Cohen,

Thank you for submitting your manuscript to PLOS ONE. After careful consideration, we feel that it has great merit but does not fully meet PLOS ONE’s publication criteria as it currently stands. Therefore, we invite you to take into account one comment made by one reviewer about a specific point of discussion, and to submit a revised version of the manuscript.

We would appreciate receiving your revised manuscript by Jun 19 2020 11:59PM. To enhance the reproducibility of your results, we recommend that if applicable you deposit your laboratory protocols in protocols.io, where a protocol can be assigned its own identifier (DOI) such that it can be cited independently in the future. For instructions see: http://journals.plos.org/plosone/s/submission-guidelines#loc-laboratory-protocols

We look forward to receiving your revised manuscript.

Kind regards,

François Blachier, PhD

Academic Editor

PLOS ONE

Journal Requirements:

2. To comply with PLOS ONE submission guidelines, in your Methods section, please provide additional information regarding the statistical analyses conducted in the study. For more information on PLOS ONE's expectations for statistical reporting, please see https://journals.plos.org/plosone/s/submission-guidelines.#loc-statistical-reporting.

Reviewers' comments:

Reviewer's Responses to Questions

**Comments to the Author**

1. Is the manuscript technically sound, and do the data support the conclusions?

Reviewer #1: Yes

Reviewer #2: Yes

2. Has the statistical analysis been performed appropriately and rigorously? 

Reviewer #1: Yes

Reviewer #2: Yes

3. Have the authors made all data underlying the findings in their manuscript fully available?

Reviewer #1: Yes

Reviewer #2: No

4. Is the manuscript presented in an intelligible fashion and written in standard English?

Reviewer #1: Yes

Reviewer #2: Yes

5. Review Comments to the Author

Reviewer #1: The authors performed liquid chromatography tandem mass spectrometry (LC/MS-MS) metabolomics profiling of human microvascular endothelial cells (HMEC-1), following short-term (48hrs) or long-term (3 weeks) hypoxia. The study has been conducted according to high technical standards and results are reported in a clear and detailed form. The article is well written and figures are presented in a comprehensible way. Overall, the aim of this study is to identify novel metabolic alterations in endothelial cells in response to hypoxia. HMECs under hypoxia cannot resemble the more complex environmental condition in which tumor-associated endothelial cells dwell (as stated by the authors too). Nevertheless, these finding are interesting and may be helpful in discovering new vulnerability points in tumor-associated endothelial cells. In conclusion, I feel to recommend this work for publication.

Reviewer #2: The authors describe changes in the profile of endothelial cell metabolic pathways in response to hypoxia, looking for potential targets to normalize tumor vasculature and improve drug delivery. The presented results are very interesting and important. The article is well written and clear. A very large number of tests were performed, including complementary tests, so the observations seem appropriate. I have no negative comments.

However, authors should discuss the results obtained considering changes in aerobic and anaerobic metabolism of endothelial cells, including alternations in oxidation of carbohydrates, fatty acids and amino acids, as well as in the functioning of endothelial mitochondria in response to hypoxia (Koziel et al., Pflugers Arch - Eur J Physiol, 2017).

6. PLOS authors have the option to publish the peer review history of their article (what does this mean?). If published, this will include your full peer review and any attached files.

Reviewer #1: No

Reviewer #2: No

---

## [Author Response · Author response to Decision Letter 0]

6 May 2020

Reviewer #1: We thank you for your positive review of our study. 

Reviewer #2: We thank you for suggesting an additional discussion of aerobic versus anaerobic metabolism, and your recommendation of a relevant article for citation. This is an excellent point. We have now included such a discussion on pages 20 and 22, and added the relevant citation. Furthermore, we have clarified that all underlying data is in fact available as Supporting Information, including full blot images as S1_raw_images.

---

## [Editor Report · Decision Letter 1]

8 May 2020

Metabolic pathway alterations in microvascular endothelial cells in response to hypoxia

PONE-D-20-09862R1

Dear Dr. Cohen,

We are pleased to inform you that your manuscript has been judged scientifically suitable for publication and will be formally accepted for publication once it complies with all outstanding technical requirements.

With kind regards and congratulation for this very nice paper,

François Blachier, PhD

Academic Editor

PLOS ONE
---

## [Editor Report · Acceptance letter]

29 Jun 2020

PONE-D-20-09862R1 

Metabolic pathway alterations in microvascular endothelial cells in response to hypoxia 

Dear Dr. Cohen:

I'm pleased to inform you that your manuscript has been deemed suitable for publication in PLOS ONE. Congratulations! Your manuscript is now with our production department. 

Kind regards, 

on behalf of

Dr. François Blachier 

Academic Editor

PLOS ONE